# Complete Genome Sequencing Revealed the Potential Application of a Novel *Weizmannia coagulans* PL-W Production with Promising Bacteriocins in Food Preservative

**DOI:** 10.3390/foods12010216

**Published:** 2023-01-03

**Authors:** Yu Wang, Zelin Gu, Shiqi Zhang, Pinglan Li

**Affiliations:** Key Laboratory of Functional Dairy, College of Food Science and Nutritional Engineering, China Agricultural University, Beijing 100083, China

**Keywords:** *Weizmannia coagulans*, bacteriocin, complete genome sequencing

## Abstract

*Weizmannia coagulans* is an important potential probiotic with dual characteristics of *Bacillus* and *Lactobacillus*. This study describes a novel *Weizmannia coagulans* PL-W with excellent antibacterial activity isolated from Mongolian traditional cheese, in which safety and probiotic potential were evaluated by complete genome sequencing. The crude bacteriocins of *W. coagulans* PL-W showed antibacterial activity against various foodborne pathogens, including *Listeria monocytogenes* CMCC 54,004, *Bacillus cereus* ATCC 14,579, and *Staphylococcus aureus* ATCC 25,923. Moreover, the crude bacteriocins have outstanding stability against pH, temperature, surfactants, and are sensitive to protease. The complete genome sequencing revealed *W. coagulans* PL-W consists of 3,666,052-base pair (bp) circular chromosomes with a GC content of 46.24% and 3485 protein-coding genes. It contains 84 tRNA, 10 23S rRNA, 10 16S rRNA, and 10 5S rRNA. In addition, no risk-related genes such as acquired antibiotic resistance genes, virulence, and pathogenic factors were identified, demonstrating that *W. coagulans* PL-W is safe to use. Furthermore, the presence of gene clusters involved in bacteriocin synthesis, adhesion-related genes, and genes contributing to acid and bile tolerance indicate that *W. coagulans* PL-W is a potential candidate probiotic. Thus, antimicrobial activity and genome characterization of *W. coagulans* PL-W demonstrate that it has extensive potential applications as a food protective culture.

## 1. Introduction

Foodborne diseases are a significant health problem for all of humanity [1]. Various antibiotics have been used for a long time to inhibit the growth of pathogenic bacteria and prevent their threat to human health. However, the overuse of antibiotics has led to increased multidrug resistance in bacteria, which is a global public health crisis that threatens our ability to treat bacterial infections [2,3]. On the other hand, several studies have shown that the human commensal microbiota can be affected by the continuous use of antibiotics [4,5]. Therefore, it is urgent to search for new antimicrobial substances, such as bacteriocin, to prevent the adverse effects of traditional antimicrobial substances. In this regard, several studies have shown that bacteriocins from probiotics which are generally recognized as safe (GRAS), could be used as a substitute for traditional antibiotics in the future [6,7].

Bacteriocins are antimicrobial peptides synthesized from ribosomes produced by bacteria, which inhibit the growth of related (narrow spectrum) or nonrelated (broad spectrum) microorganisms [8]. At present, the bacteriocins from microbials approved for application in food additives are mainly from Nisin and Pediocin PA-1 produced by lactic acid bacteria (LAB) [9,10]. Bacteriocins predominantly exert their antibacterial activity by influencing gene and protein replication, pore formation, and membrane permeabilization [11,12], which could prevent target bacteria from evolving into corresponding drug-resistant strains, making them a potential alternative to antibiotics. It is widely known that the genus *Bacillus* is a rich source of bacteriocins or bacteriocin-like inhibitory substances (BLIS) [13,14,15]. Compared with most LAB, bacteriocins produced by *Bacillus* have broader inhibition spectra and may include Gram-positive bacteria, Gram-negative bacteria, or fungi, some of which are pathogenic for humans [16]. Consequently, bacteriocins from *Bacillus* are increasingly becoming more critical and have attracted more and more interest from researchers.

*W. coagulans* is a high-temperature-resistant spore-forming bacterium with probiotic activity. It has received extensive attention since it contains the properties both of *Bacillus* and *Lactobacillus* [17]. In 2012, the Food and Drug Administration approved *W. coagulans* as safe to use in food [18]. Several studies have found that some *W. coagulans* possess good antibacterial activity by producing bacteriocin [19]. For instance, lactosporin produced by *W. coagulans* ATCC 7050 has been reported to inhibit pathogens [20]. The bacteriocin produced by *W. coagulans* can inhibit the growth of spoilage bacteria and improve the preservation of large yellow croaker [21]. *W. coagulans* BDU3 produces a new 1.4 kDa bacteriocin with antimicrobial activity against foodborne pathogens [22]. Therefore, *W. coagulans* are expected to become a new star bacterium for food preservatives. There are three conventional strategies for using bacteriocins in the food industry: pure bacteriocin, bacteriocin-containing fermentates, and bacteriocin-producing live cells. It is noteworthy that the strains used to produce bacteriocin must be safe. Thus, assessing the safety and probiotic properties of the strains is an essential step for use in food products [23,24]. For instance, Sreenadh et al. evaluated the probiotics, safety, and technology of *W. coagulans* S-31,876 through a number of in vitro experiments to explore its potential applications [25]. High-throughput sequencing enables the evaluation of the properties of strains at the genomic level, including their genetic, safety, and metabolic profiles. Previously, Aulitto et al. used comparative genomics to focus on the biotransformation and defense ability of *W. coagulans* against the external environment [26]. In this research, the *W. coagulans* PL-W with antibacterial activity was identified from Mongolian traditional cheese, and its complete genome was confirmed. Assessment of the genome indicates that *W. coagulans* PL-W may be a safe strain with probiotic properties to use and promote future research and development of the organism in food preservation. In addition, the crude bacteriocin characteristics produced by *W. coagulans* PL-W were evaluated to provide the theoretical basis for its potential application as a food preservative.

## 2. Materials and Methods

### 2.1. Samples and Bacterial Culture Conditions

*W. coagulans* PL-W were cultured in Man Rogosa and Sharpe (MRS) medium at 45 °C. The indicator strain, *L. monocytogenes* CMCC 54,004, was cultured in TSYEB medium at 37 °C. The medium used for the culture of other bacteria is shown in Table 1. All bacteria used in this research were stored at −80 °C in a suitable culture medium containing 25% (*v*/*v*) glycerol.

### 2.2. Isolation of Antimicrobial Substance-Producing W. coagulans and Crude Antimicrobial Substance Preparation

The method of isolation of *W. coagulans* was based on the previous method with some modifications [27]. Firstly, Mongolian traditional cheese was homogenized in sterilized water and then heated to 80 °C for 10 min. Subsequently, samples were diluted in a gradient and then spread individually on MRS agar with 2 g/L CaCO_3_ and incubated at 45 °C for 48 h. Bacterial colonies that showed clear circles on the plates containing CaCO_3_ were individually picked and then inoculated on MRS broth medium and incubated at 45 °C under the shaking condition (200 rpm) for 48 h.

The cell-free supernatant (CFS) was obtained by centrifugation at 10,000× *g* for 10 min, and then ammonium sulfate was slowly added to the CFS to 80% saturation by stirring and at 4 °C overnight. To collect the crude antimicrobial substance, the mixture was centrifuged at 10,000× *g* for 20 min at 4 °C, and then the precipitate was resuspended in 1 mL of PBS (pH 6.8). The activity of crude antimicrobial substance against *L. monocytogenes* CMCC 54,004 was determined by the agar well diffusion method. Strains with strong antimicrobial activity against the *L. monocytogenes* CMCC 54,004 were selected and identified by Gram staining, lactate production capacity, catalase and oxidase activities, ability to grow at different temperatures (45–60 °C), NaCl concentrations (1–5%), and various sugars. The 16S rDNA sequencing was performed for genotype identification as described previously [28]. Subsequently, the MEGA 7.0 software was used to analyze the phylogenetics of strains.

### 2.3. Characteristics of Antimicrobial Substance Production in W. coagulans PL-W

#### 2.3.1. Kinetics of Growth and Crude Antimicrobial Substance Production in *W. coagulans* PL-W

*W. coagulans* PL-W was grown in 30 mL of MRS medium at 45 °C for 48 h and then inoculated into MRS medium at a 2% inoculum (*v*/*v*). The cell density at 600 nm was determined every 4 h while CFS was collected to obtain the crude antimicrobial substance and assess antibacterial activity by the method described in Section 2.2 [28].

#### 2.3.2. Physicochemical Properties of Crude Antimicrobial Substance

The stability of the crude antimicrobial substance was examined by detecting the changes in antibacterial activity against *L. monocytogenes* CMCC 54,004 under different conditions [29]. The enzyme stability of the crude antimicrobial substance was evaluated by incubation with lipase, α-amylase, proteinase K, neutral protease, flavor enzyme, trypsin, and pepsin for 2 h under the optimal reaction conditions for each enzyme, while the final concentration of each enzyme was 1 mg/mL. The pH of the crude antimicrobial substance was adjusted from 2 to 10 with 2 M HCL and 2 M NaOH to detect the change in its antibacterial activity. The thermal stability of the crude antimicrobial substance was tested by incubation at 4–121 °C for 10 and 30 min, respectively. To test the crude antimicrobial substance stability with the surfactant, the crude antimicrobial substance was incubated with various chemical reagents 1% (*v*/*v*), including Tween 80, EDTA, urea, and SDS at 37 °C for 2 h. The antimicrobial activity of the untreated antimicrobial substance was measured and used as the positive control.

#### 2.3.3. Antimicrobial Spectrum Assay of Crude Bacteriocins

To investigate the antimicrobial spectrum of *W. coagulans* PL-W, the antimicrobial activity of the crude bacteriocins of *W. coagulans* PL-W against a range of indicator strains, including food spoilage bacteria and food-borne pathogens (Table 1), was determined by using the pour plate method described by An et al. [30]. The minimal inhibitory concentration (MIC) and minimal bactericidal concentration (MBC) of the crude bacteriocin were calculated by observing the growth of *L. monocytogenes* CMCC 54004 mixed with various concentrations of the crude bacteriocins [31].

#### 2.3.4. Purification of Bacteriocin

To identify the bacteriocin, firstly, ultrafiltration tubes of 10 KD and 3 KD were used to isolate the crude bacteriocin. The fractions containing proteins larger than 10 KD, between 3–10 KD, and smaller than 3 KD were tested for antibacterial activity. Then, the active fractions were loaded onto a C18 reverse-phase column (5 µm, 4.6 × 250 mm, Agilent, Santa Clara, CA, USA), connected to a reverse-phase high-performance liquid chromatography (RP-HPLC) system, and eluted at 0.5 mL/min flow rate by a linear gradient elution with 95% water–acetonitrile (5–95%) containing 0.1% trifluoroacetic acid (TFA) for 30 min. The different peaks were collected at an absorbance of 280 nm, and then concentrated using 1 KD ultrafiltration tubes for antibacterial activity evaluation. Using Tricine-SDS-PAGE (16.5% separated and 4% concentrated gel), the range of molecular mass of the collected active fractions was analyzed.

### 2.4. Genome Sequencing, Assembly, Annotation, and Classification

*W. coagulans* PL-W grown to mid-logarithmic phase were collected and genomic DNA were extracted using the Bacterial Genomic DNA Isolation kit (TianGen, Beijing, China) according to the kit instructions. The PromethION platform was used to sequence the *W. coagulans* PL-W genome. Subsequently, the sequences were mix assembled with Unicycler and corrected with Pilon (parameter: OFF). After removing the redundant part, the Circlator (parameter: fixStart) was used to move the origin of the sequence to the replication start site of the genome to obtain the final genome sequence. The encoding gene was predicted with prodigal, tRNA, rRNA, and other ncRNAs were predicted using trnascan-SE. Interproscan was used for annotation of genome-encoded proteins. BlastP was used to align the encoded proteins to KEGG, RefSeq, and COG databases, and the best results with alignment coverage greater than 30% were retained as annotation results. The circos was used to draw the nuclear genome circle map.

Using *W. coagulans* PL-W as a reference strain, digital DNA-DNA hybridization was performed using an online tool (http://ggdc.dsmz.de/, (accessed on 5 September 2022)). The *W. coagulans* PL-W genome in the form of FASTA was submitted to the Type (Strain) Genome Server (TYGS) and the genome was compared with the TYGS database.

### 2.5. Prediction of the Safety of W. coagulans PL-W

The Comprehensive Antibiotic Research Database (CARD) and Resistance Gene Identification Tool (RGI) were used to analyze the presence of drug-resistance genes in *W. coagulans* PL-W. The VFDB webserver was used to predict putative virulence factors and the pathogen finder webserver was used to predict bacterial pathogenicity.

### 2.6. Prediction of the Probiotic Characteristics of W. coagulans PL-W

The Hidden Markov model (HMM) was used to detect genes related to acid and bile tolerance in the genome of *W. coagulans* PL-W and various proteins related to adhesion and aggregation were searched in the genome annotation data. AntiSMASH5 and BAGEL4 were used to predict non-ribosomal synthetic secondary metabolites (NRPS) and bacteriocin synthesis gene clusters in the *W. coagulans* PL-W genome, respectively.

### 2.7. Statistical Analysis

Three parallel groups were set up for each group of experiments. GraphPad Prism software was used for statistical analysis of the experimental data by a one-way ANOVA, and values of *p* < 0.05 were considered statistically significant.

## 3. Results and Discussion

### 3.1. Screening and Identification of Antimicrobial Substance-Producing Strains

Bacteriocins are a potential alternative to antibiotics since they are safe and it is not easy to produce antibiotic resistance to indicator strains [32]. Bacteriocins produced by *Bacillus* have a broad inhibitory spectrum, which makes them of excellent research significance [16]. The present study isolated 243 potential lactate-producing strains from Mongolian traditional cheese. Among these strains, strain PL-W possessed the highest antibacterial activity to indicator strains (*L. monocytogenes* CMCC 54,004) and was selected as the target strain. It was characterized as Gram-positive, spore-forming, rod-shaped, lactate produced, and positive for catalase, indole, and the Voges–Proskauer test, and grew in medium containing 5% sodium chloride and fermented sucrose, glucose, lactose, and arabinose. The phylogenetic analysis according to the strain 16S rRNA sequence showed that the selected strain belonged to the same evolutionary branch as *W. coagulans* and shared 100% support degree with *W. coagulans* 683, *W. coagulans* DSM1 ATCC 7050, and *W. coagulans* NBRC 12,583 (Appendix A). Thus, we classified the strain PL-W as *W. coagulans* PL-W.

### 3.2. Crude Antimicrobial Substance Production Properties of W. coagulans PL-W

#### 3.2.1. Kinetics of Growth and Crude Antimicrobial Substance Production in *W. coagulans* PL-W

Kinetics of crude antimicrobial substance production of *W. coagulans* PL-W cultured in MRS broth at 45 °C with shaking are presented in Figure 1. The inhibition zone size of crude antimicrobial substance against *L. monocytogenes* 54,004 was used to detect its production. The result showed that *W. coagulans* PL-W entered the stable phase after 24 h of culture. Meanwhile, the inhibition zone against *L. monocytogenes* 54,004 was detected and reached the largest at 32 h, indicating the production of the crude antimicrobial substance may reach the maximum mass. The antibacterial activity of the crude antimicrobial starts to decrease slightly after 32 h, probably due to degradation by other substances secreted by *W. coagulans* PL-W. Thus, the crude antimicrobial substance obtained from *W. coagulans* PL-W was cultured at 32 h for further characterization.

#### 3.2.2. Characterization of Crude Antimicrobial Substance

Bacteriocins are antimicrobial peptides synthesized by ribosomes, produced by LAB, which have drawn wide attention owing to their characterization as food antiseptics. For example, Nisin, produced by *Lactococcus lactis*, has been approved by the Food and Drug Administration (FDA) and used for food preservation since 1950 [33]. Pediocin, bavaricin, leucocin, and sakacin are other bacteriocins produced by LAB that have been approved for use in food in some countries [34,35]. Although some *Bacillus* species are involved in various food fermentation processes, none of the *Bacillus* bacteriocins have been approved as food preservatives [36,37]. *W. coagulans* has been reported as safe by the FDA and the European Union Food Safety Authority (EFSA), which was classified as *Bacillus coagulans* due to its *Bacillus* characteristics. Therefore, there are further opportunities for *W. coagulans* bacteriocins to prove useful for food preservation. In this study, the antimicrobial substance of *W. coagulans* PL-W was treated with different enzymes to determine its nature. The antimicrobial activity against *L. monocytogenes* CMCC 54,004 indicated that the antimicrobial substance remained stable after lipase and α-amylase treatment. On the contrary, the antimicrobial substance was sensitive to proteinase K, indicating that the antimicrobial substance is a bacteriocin (Figure 2A). Furthermore, we examined the stability of the crude bacteriocin of *W. coagulans* PL-W under different conditions, including pH, temperature, and surfactants, to predict its application in the food industry. In the test of pH stability, the antibacterial activity of crude bacteriocins remained above 90% at pH 2–8, which indicates that the crude bacteriocins still have specific activity under acidic, neutral, and weakly alkaline conditions (Figure 2B). Therefore, bacteriocins produced by *W. coagulans* PL-W have the potential to be applied to acidic, neutral, and weakly alkaline foods. The antibacterial activities of the bacteriocins were nearly not altered under 4 °C to 40 °C, indicating that bacteriocins may have storage stability. Although the activity decreased after treatment at 60–100 °C, the crude bacteriocins show more than 70% activity making them valuable for heat-processed foods (Figure 2C). The stability of bacteriocins against surfactants facilitates their use in emulsifying food. Our study showed that the crude bacteriocins had little influence on antibacterial activity after treatment with different surfactants, suggesting that bacteriocins could be used in emulsified food (Figure 2D).

#### 3.2.3. Antibacterial Spectrum of Crude Bacteriocins

The antibacterial spectrum of crude bacteriocins was determined by assaying for the antimicrobial activity against several indicator strains. The crude bacteriocins displayed a wide range of the antibacterial spectrum, including Gram-positive and Gram-negative bacteria (Table 1). Significantly, some of them are common spoilage bacteria in the food industry, such as *L. monocytogenes* CMCC 54,004, *S. aureus* ATCC 25,923, and *B. cereus* ATCC 14,579. The result of this study suggests that *W. coagulans* PL-W and its bacteriocin products have great utilization potential in food preservation.

#### 3.2.4. Purification of Bacteriocin

The crude antimicrobial substances were separated into different fractions using 10 KD and 3 KD ultrafiltration tubes. The result showed that only fractions with molecular mass between 3–10 kDa exhibited antibacterial activity (Figure 3A). Therefore, it was presumed that the molecular mass of the bacteriocin was among 3–10 kDa. Further purification of the active fraction using a C18 column showed that only the third peak was able to inhibit the growth of the indicator strain (Figure 3B). Tricine-SDS-PAGE analysis of the molecular mass of peak 3 displayed a single band around 7 kDa (Figure 3C). Consequently, we conclude that *W. coagulans* PL-W expressed a bacteriocin with a molecular mass near 7 kDa.

### 3.3. General Genome Features of W. coagulans PL-W

*W. coagulans* are widely used as probiotics in the food industry, medicine, and animal breeding [17]. Genetic analyses assist humans in finding potential probiotic strains with genetically encoded properties such as bile acid resistance, epithelial adhesion, and bacteriocins, but also determine whether a strain is safe to use from a genetic perspective [26,38]. There is currently insufficient information on the probiotic effects of *W. coagulans* on the genetic basis. Therefore, the present study aims to elucidate the safety and probiotic properties of *W. coagulans* PL-W and explore its potential application in food preservation through genomic analysis. Whole genome sequencing was performed using the PromethION platform. A total of 3,554,753,206 raw reads were used for genome assembly with the unicycler assembler, version 0.4.8 (Ryan R. Wick, Victoria, Australia). The complete genome of *W. coagulans* PL-W consists of 2 contigs of 3,666,052 bp with a GC content of 46.24%. None of the plasmid sequences were validated with the Plasmid Finder 2 tool. The protein-coding genes, ribosomal RNA, and transfer RNA of the genome are shown in Table 2. The functional classification of protein was analyzed in Table 3 through the COG database. KEGG and GO databases were also used functionally to annotate protein-coding genes (Appendix A). The complete genome information of *W. coagulans* PL-W is shown in Figure 4.

### 3.4. Taxonomic Classification and Phylogeny

Phylogenetic analysis at the genome level has contributed to evaluating the diversity of *W. coagulans* species and the taxonomic status of the species. Type (Strain) Genome Server (TYGS) is a high-throughput platform that is the most advanced genome-based classification platform [39]. The results of TYGS analysis demonstrated that *W. coagulans* PL-W is similar to *W. coagulans* ATCC7050 and *W. coagulans* DSM1 (Figure 5). Digital DNA–DNA hybridization (DDH) of the *W. coagulans* PL-W genome was performed using the publicly available genome sequences of three *W. coagulans* (DSM1, 2–6, ATCC7050, and XZL9) using the Genome–Genome Distance Calculator (GGDC) 3.0 server [40]. The results indicated that *W. coagulans* PL-W shares 94.21% and 85.32% similarity with *W. coagulans* XZL9 and *W. coagulans* 2–6, proving that *W. coagulans* PL-W is a new strain of *Weizmannia*. Using JSpeciesWS to calculate average nucleotide identity (ANI) using the method described previously revealed a 98.23% similarity of *W. coagulans* PL-W with *W. coagulans* XZL9 [41]. Thus, *W. coagulans* PL-W was identified as a member of the *W. coagulans* species based on >70% similarity in DDH and ~95% or higher ANI with a similar reference strain.

### 3.5. Safety Assessment of W. coagulans PL-W

Safety assessment is an essential process for selecting probiotics. According to the recommendations of Qualified Safety Presumption (QPS) approved by the European Food Safety Authority (EFSA), the presence and potential mobility of antibiotic resistance genes should be considered when selecting new probiotics since the increasing resistance of bacteria to antibiotics poses a great threat to human health [42]. The CARD database contains bacterial drug resistance genes from different environmental sources (such as the gut, domestic wastewater, rivers, etc.) and their annotated information such as their resistance spectrum, mechanism of action, ontology, COG, and CDD [43]. Analysis of the *W. coagulans* PL-W genome from the CARD database showed that *W. coagulans* PL-W does not contain any antibiotic resistance genes or acquired antimicrobial genes detected. Similarly, phenotypic studies of *W. coagulans* PL-W sensitivity to the antibiotics were all below the cut-off values mentioned in the EFSA prescribed standards (Table 4).

The Virulence Factor Database (VFDB) collects information on the virulence factors of bacterial pathogens [44]. The existence of any virulence genes was not detected on the *W. coagulans* PL-W genome using the VFDB service. At the same time, we found that *W. coagulans* PL-W did not exhibit hemolytic zones on the Columbia Agar plates with sheep blood (Appendix A). Based on the above analysis, we promoted that *W. coagulans* PL-W was presumed to be a safe strain.

### 3.6. Assessment of Probiotic Properties

The ability to endure harsh conditions in the gastrointestinal tract is one of the criteria for selecting probiotics [45]. A large number of genes involved in acid and bile salt tolerance have been identified in the *W. coagulans* PL-W genome. ATP synthase (F1–F0-ATPase), a group of proteins mainly involved in acid resistance, are known to effectively pump protons out of the cell by hydrolyzing ATP, maintaining a low proton concentration inside the cell, thereby improving acid tolerance and thus maintaining pH homeostasis. There are eight coding F0F1 ATP synthase genes (atpC, atpD, atpG, atpA, atpH, atpF, atpE, and atpB) that were identified in the *W. coagulans* PL-W genome. In addition, two (Na+/H+) transport protein genes, one (H+/Cl−) antitransporter gene, ClcA, and one (Ca^2+^/H+) antitransporter gene, chaA, were detected in the *W. coagulans* PL-W genome, which has been shown to play critical roles in pH and homeostasis of cells [46,47]. Bile salt hydrolases belong to the family of glycine hydrolase, which act by binding bile salts to counteract the harmful effects of bile. A gene-encoding cholylglycine hydrolase was identified in the *W. coagulans* PL-W genome. Two sodium bile acid symporter family genes were also identified, and the presence of these genes was beneficial for the bile tolerance of *W. coagulans* PL-W [48]. The survival rate of *W. coagulans* PL-W at 1% bile salt concentration and pH 2.0 further validated the genomic data.

Adhesion to the intestinal epithelium has been considered an important probiotic property since adhesion can promote intestinal colonization while competitively excluding harmful pathogens [49,50]. We searched the genes of adhesion, colonization, mucin binding, flagellar hook, and fibrinogen/fibronectin binding from annotated data. There were 11 genes found to encode adhesion-related proteins in *W. coagulans* PL-W, including hook-associated protein FlgK and FlgL, flagellar filament capping protein FliD, flagellar hook–basal body complex protein FliE, flagellar hook–length control protein FliK, flagellar hook assembly protein FlgD, flagellar basal body rod protein FlgF, fibronectin/fibrinogen-binding protein (FbpA), DUF817 domains (fibronectin/fibrinogen binding protein), and segregation and condensation protein (scpA and scpB) (fibronectin-binding protein). Flagella can act directly as adhesins and play a key role in colonization by facilitating bacterial motility [51]. Thus, several proteins related to flagellar formation were detected in *W. coagulans* PL-W, indicating that *W. coagulans* PL-W may have good motility. Fibronectin is an important multidomain glycoprotein with various adhesion properties and serves as an essential link between cells and their extracellular matrix [52]. Additionally, some similar types of adhesion proteins have been identified in genome-wide analyses of *W. coagulans* S-Lac and *W. coagulans* GBI-30, helping the *W. coagulans* to colonize in host intestines and act as probiotic [50].

### 3.7. Antimicrobial Compound Gene Prediction and Validation

The characteristics of synthesizing antimicrobial compounds play an important role in the competition of survival exclusion when probiotics are colonized in the gut [11]. Genome analysis is expected to pave the way to finding new antimicrobial compounds and understanding the mechanisms of antimicrobial compound production in bacteria [53]. Therefore, BAGEL 4.0 and AntiSMASH 5.0 were used to predict antimicrobial compounds in *W. coagulans* PL-W. Polyketides (PKs)-T3PK3 biosynthesis, betalactone, and two RiPP-like compounds were identified by antiSMASH. The two RiPP-like compounds were both identified as Circularin A and Amylocyclicin by BAGEL 4.0 (Figure 6). Circularin A and Amylocyclicin are typically produced as a propeptide after the leader peptide is cleaved, and then ligation between the N- and C-termini results in cyclic antimicrobial peptides. Circularin A produced by *Clostridium beijerinckii* ATCC 25,752 has been reported to have broad-spectrum antibacterial activity [54], and Amylocyclicin produced by *Bacillus amyloliticus* FZB42 has a wide range of antibacterial activity against Gram-positive bacteria [55]. It is worth noting that the putative Circularin A and Amylocyclicin synthetic gene clusters of *W. coagulans* PL-W were not found to be similar to any known cluster by antiSMASH servicer. The ABC transporter may be responsible for the transport of mature peptides to protect itself from bacteriocin attack [56], which was both found on the putative Circularin A and Amylocyclicin synthesis gene cluster, suggesting that these two bacteriocins may be secreted in the form of ABC transport. A gene annotated with CirC protein that had a modification function related to bacteriocin cyclization was detected in the Circularin A synthesis gene cluster [57]. Meanwhile, a histidine kinase gene was detected on the putative Amylocyclicin synthesis gene cluster, which predicted that Amylocyclicin synthesis might be regulated by quorum sensing [58]. A transcriptional regulatory protein and competence regulatory protein may be involved in the transcription and cyclization of Amylocyclicin.

Tricine-SDS-PAGE analysis of the bacteriocins indicated that its molecular mass was near 7 kDa (Figure 3C); combining the genomic information, we concluded the bacteriocin extracted from *W. coagulans* PL-W was Circular A and the entire amino acid sequence was MGLFHVASKFHVSAGIASGVVTAVLHAGTIASIIGAVTVVMSGGVDAILDMGWTAFIAEVKHLAKEYGKKRAIAW. Although the genome was confirmed for the presence of the gene of Amylocyclicin, purification of crude bacteriocin did not find it. Thus, we speculate that Amylocyclicin may be expressed only under certain specific growth conditions, or Amylocyclicin did not show inhibitory activity against the *L. monocytogenes* CMCC 54,004 that we used in the purification process. In this regard, it may be possible to use heterologous expression methods in the future to detect the antimicrobial ability of Amylocyclicin or attempt to use other bacteria rather than *L. monocytogenes* as the indicator strain when purifying the bacteriocins. In general, the expression of Circularin A may provide a competitive advantage for *W. coagulans* PL-W.

## 4. Conclusions

The current study identifies a bacteriocins-producing *W. coagulans* PL-W from Mongolian traditional cheese. The crude bacteriocin extracted from *W. coagulans* PL-W by 80% ammonium sulfate not only exhibited a broad antimicrobial spectrum including a Gram-positive bacterium and a Gram-negative bacterium, some of which are common foodborne pathogens, but has good stability to surfactant, heat, and pH. Genome analysis indicated that *W. coagulans* PL-W is a safe strain to use. In addition, the strain harbors genes encoding two bacteriocins which might ensure *W. coagulans* PL-W has excellent antibacterial ability. Research of the crude bacteriocin characteristics and genomic sequencing suggested that *W. coagulans* PL-W is a safe candidate for controlling foodborne pathogens and is promising for use in the food industry. Moreover, the whole genomic results will contribute to further in vitro and in vivo investigations of *W. coagulans* PL-W to prospect its application as a probiotic.

## Figures and Tables

**Figure 1 foods-12-00216-f001:**
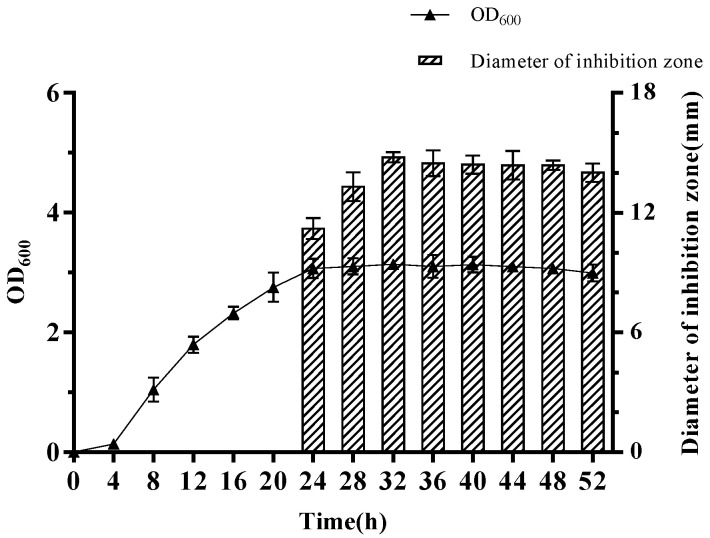
Growth and dynamics of antimicrobial substance production by *W. coagulans* PL-W.

**Figure 2 foods-12-00216-f002:**
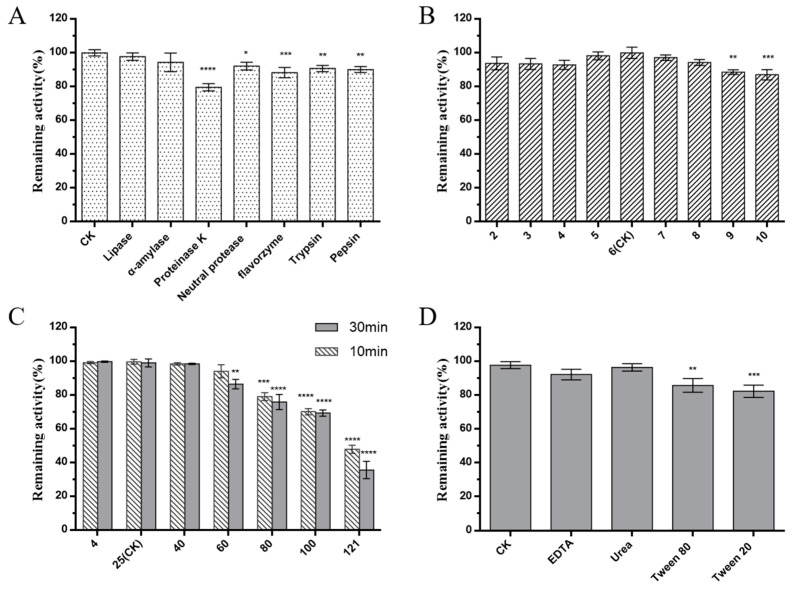
Stability of crude bacteriocins. (**A**) enzyme; (**B**) pH; (**C**) temperature; and (**D**) surfactant. The experiment was repeated three times. (*n* = 3, * *p* < 0.05; ** *p* < 0.01; *** *p* < 0.001; **** *p* < 0.0001).

**Figure 3 foods-12-00216-f003:**
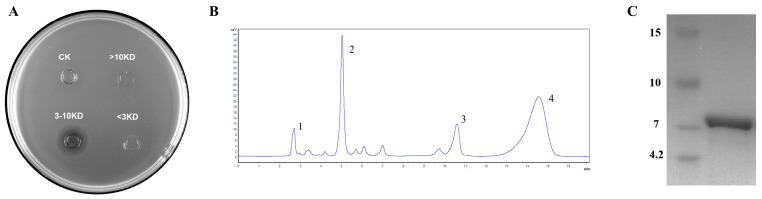
Purification of Bacteriocin. (**A**) Ultrafiltration tubes separate the active fractions. (**B**) RP-HPLC process of purification. (**C**) Tricine–SDS–PAGE of activity fraction.

**Figure 4 foods-12-00216-f004:**
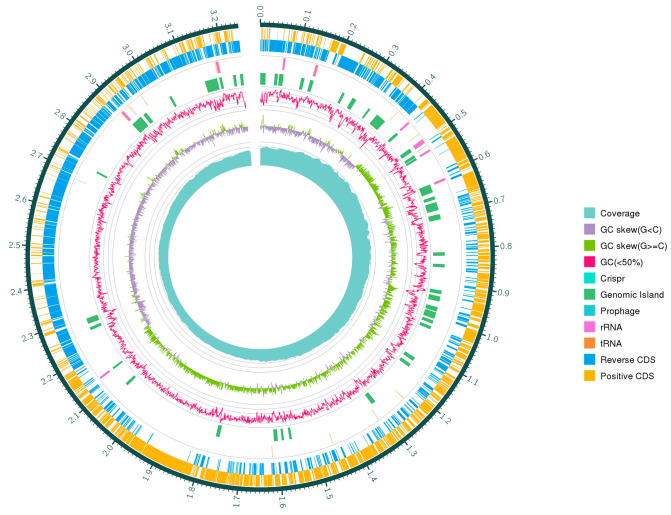
Circular genome map of *W. coagulans* PL-W. From outside to center, ring 1: encoding gene (positive CDS); ring 2: encoding gene (reverse CDS); ring 3: tRNA and rRNA; ring 4: CRISPR, prophage and genomic island; ring 5: GC content; ring 6: GC-skew; and ring 7: coverage.

**Figure 5 foods-12-00216-f005:**
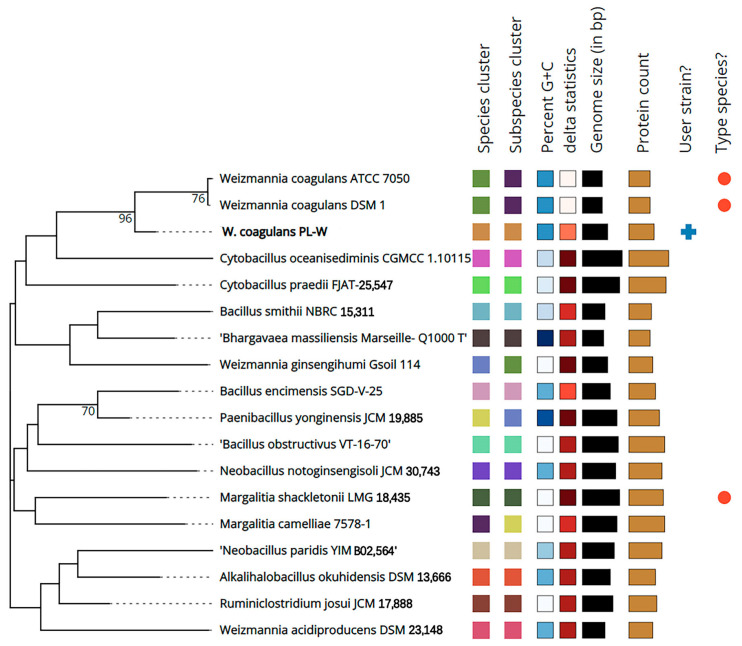
Phylogenetic analysis of *W. coagulans* PL-W whole genome by TYGS.

**Figure 6 foods-12-00216-f006:**
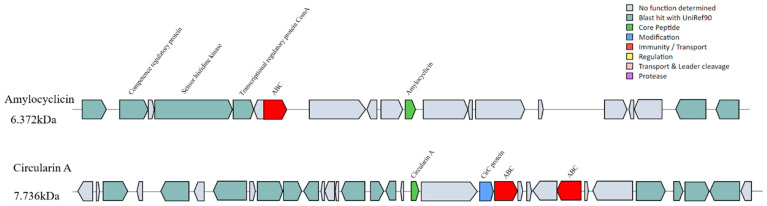
Bacteriocin synthesis gene cluster predicted by BAGEL4. Two cluster-encoding genes related to Circularin A and Amylocyclicin were identified in the *W. coagulans* PL-W genome.

**Table 1 foods-12-00216-t001:** Antibacterial spectrum of the crude bacteriocin of *W. coagulans* PL-W.

Indicator Strain	Source	Media	Activity (mm) ^a^	MIC (µg/mL)	MBC (µg/mL)
Gram-positive bacteria					
*Listeria monocytogenes* CMCC 54,004	Lab	TSYEB	+++	55.32	221.28
*Staphylococcus aureus* ATCC 25,923	Lab	TSB	+	221.28	442.56
*Bacillus cereus* ATCC 14,579	Lab	LB	+	221.28	442.56
*Bacillus subtilis*	Lab	LB	++	110.64	442.56
*Bacillus licheniformis*	Lab	LB	+++	55.32	442.56
*Bacillus amyloliquefaciens*	Lab	LB	+	221.28	885.13
*Lactobacillus plantarum*	Lab	MRS	+++	55.32	221.28
*lactococcus lactis* MG1363	Lab	MRS	-		
*Enterococcus faecalis*	Lab	MRS	-		
*Lactobacillus bulgaricus*	Lab	MRS	-		
*Lactococcus lactis* NZ9000	Lab	MRS	-		
Gram-negative bacteria					
*Escherichia coli* BL21	Lab	LB	-		
*Escherichia coli* BW25113	Lab	LB	-		
*Pseudomonas aeruginosa*	Lab	LB	+	442.56	885.13

^a^ Diameter of inhibition zone: +++, ≥15 mm; ++, ≥12 mm; +, >8 mm; -, no inhibition zone. The diameter of the hole was 8 mm. The diameter of the circular hole was 8 mm.

**Table 2 foods-12-00216-t002:** General genome attributes of *W. coagulans* PL-W.

Attributes	*W. coagulans* PL-W
Genome size (bp)	3,666,052
No. of contigs	2
GC content %	46.24
Coding DNA sequence (CDS)	3485
rRNAs	30
tRNAs	84

**Table 3 foods-12-00216-t003:** COG categories of coding proteins in *W. coagulans* PL-W.

COG Class	Name	Count	Proportion (%)
C	Energy production and conversion	143	4.81
D	Cell cycle control, cell division, chromosome partitioning	180	6.05
E	Amino acid transport and metabolism	265	8.90
F	Nucleotide transport and metabolism	96	3.23
G	Carbohydrate transport and metabolism	254	8.53
H	Coenzyme transport and metabolism	156	5.24
I	Lipid transport and metabolism	169	5.68
J	Translation, ribosomal structure, and biogenesis	212	7.12
K	Transcription	221	7.43
L	Replication, recombination, and repair	127	4.27
M	Cell wall/membrane/envelope biogenesis	127	4.27
N	Cell motility	49	1.65
O	Post-translational modification, protein turnover, chaperones	119	4.00
P	Inorganic ion transport and metabolism	133	4.47
Q	Secondary metabolite biosynthesis, transport and catabolism	41	1.38
R	General function prediction only	184	6.18
S	Function unknown	122	4.10
T	Signal transduction mechanisms	165	5.54
U	Intracellular trafficking, secretion, and vesicular transport	29	0.97
V	Defense mechanisms	94	3.16
W	Extracellular structures	9	0.30
X	Mobilome: prophages and transposons	75	2.52
Z	Cytoskeleton	6	0.20

**Table 4 foods-12-00216-t004:** *W. coagulans* PL-W sensitivity to antibiotics.

Antibiotics	MIC (µg/mL)	MIC Cut-off Values (µg/mL)	Interpretation
Clindamycin	0.125	4	S
Gentamicin	0.25	4	S
Streptomycin	0.25	8	S
Vancomycin	0.5	4	S
Erythromycin	2	4	S
Kanamycin	0.25	8	S
Tetracycline	0.25	8	S
Chloramphenicol	4	8	S

S—susceptible; R—resistant.

## Data Availability

Not applicable.

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
