# Peer review of "Complete Genome Sequencing Revealed the Potential Application of a Novel Weizmannia coagulans PL-W Production with Promising Bacteriocins in Food Preservative"

_foods, 2023, doi:10.3390/foods12010216_

Round 1
Reviewer 1 Report
Dear authors,
I read with great attention your manuscript that I find well written and the subject very interesting.
I suggest you some corrections. Plaese take the time to do them.
1. In the manuscript, I can not find the minimal inhibitory concentration (MIC) and minimal bactericidal concentration (MBC) of the antibacterial substance of W. coagulans PL-W.
2. Line 91-93: The authors found that the bacteriocin of W. coagulans PL-W can be roughly extracted when the ammonium sulfate concentration exceeded 80 %. In the process of crude extraction, there must be some loss of bacteriocin. What is the loss rate and extraction rate of bacteriocin? What is the proportion of purity bacteriocin in the crude extract of bacteriocin? The author did not provide relevant experimental results. In addition, SDS-PAGE of crude extract of bacteriocin need to be provided.
3. Line 96-98: In combination with analysis of 16S rDNA sequences, biochemical and physiological properties, the strains can be accurately determined. It is recommended that the authors add the physiological identification index of the W. coagulans PL-W in this paper.
4. Line 177-180: "… reached the largest at 36 h, indicating the production of the antimicrobial substance … cultured at 36 h for further characterization." There seems to be no difference in the antibacterial effect between 32 h and 36 h. However, the authors did not prove that the antibacterial substances produced by the bacteria did not increase after 36 h. Please give the corresponding explanation that why 36 h is the most appropriate incubation time.
5. Line 182: "Growth and dynamics of antimicrobial substance production by W. coagulans PL-W." It is suggested that the authors should supplement the experiment, and accurately determine the amount of bacteriocin produced by W. coagulans PL-W according to the relevant literature, so as to obtain the correct dynamic law of antibacterial substance production.
6. Figure 2: (1) It is recommended to adjust the picture-ordering of Figure 2 so that the result analysis is consistent with the picture order. In addition, the font and size of the coordinate axis information should be unified. (2) What is the evidence for the temperature gradient in the temperature stability of bacteriocins? From the Figure 2-A, the temperature stability of bacteriocins decreased significantly after treatment at 60 °C. However, at lower temperatures, whether the temperature stability of bacteriocins decreased or increased, it should be explained.
Reviewer 2 Report
The work demonstrates the characterization and evaluation of safety, antimicrobial activity and probiotic properties of Weizmannia coagulans PL-W using WGS data. The manuscript is well organized, sufficiently prepared and written with appropriate English language level. In my humble opinion, however, several minor points as indicated below should be clarified and revised.
· Line 23: I would suggest replacing “food preservative” by “food biocontrol or protective culture”
· Line 85-88: Reference for the isolation method should be mentioned.
· Line 107-119: Appropriate references should be also mentioned.
· Line 112: Reconsider the word “protease stability” since various group of enzymes, i.e. lipase and amylase, were also tested in this study.
· Line 122: Please indicate the indicator strains tested here.
· Line 164: Why L. monocytogenes was selected as an indicator strain at this primary screening step?
· Line 170: W. coagulans has been recognized as GRAS from the food safety lists such as EFSA (QPS) and IDF/EFFCA microbial food cultures or not?
· Line 277-282: This data should be also addressed as supplementary materials.
· Line 284-317: Do the authors think that it is necessary to confirm these probiotic potentials using in vitro assays? I would recommend discussing also on these confirmation issues.
Author Response
Response to Reviewer 2 Comments
Dear reviewer:
Thank you for your decision and constructive comments on our manuscript. We have carefully considered your suggestion and make some changes. We have tried our best to improve and made some changes in the manuscript.
We are grateful for your comments and have addressed them as follows:
Point 1: Line 23: I would suggest replacing “food preservative” by “food biocontrol or protective culture”.
Response 1: Many thanks for your suggestions. We have replaced “food preservative” with “food protective culture”. Please see line 23 of the revised manuscript.
Point 2: Line 85-88: Reference for the isolation method should be mentioned.
Response 2: We thank the reviewer for pointing this out, and a reference for the isolation method has been added.
The changed contexts read:
Line 85-87: The method of isolation of W. coagulans was based on the previous method with some modifications[25]. Firstly, Mongolian Traditional cheese was homogenized in sterilized water and then heated to 80℃ for 10 min.
References are as follows:
[25] Abdhul, K.; Ganesh, M.; Shanmughapriya, S.; Kanagavel, M.; Anbarasu, K.; Natarajaseenivasan, K. Antioxidant activity of exopolysaccharide from probiotic strain Enterococcus faecium (BDU7) from Ngari. Int J Biol Macromol. 2014, 70, 450-454. doi: 10.1016/j.ijbiomac.2014.07.026.
Point 3: Line 107-119: Appropriate references should be also mentioned.
Response 3: We thank the reviewer for pointing this out, we have added a reference to lines 107-119.
References are as follows:
[27] Wang, Y.; Qin, Y.; Xie, Q.; Zhang, Y.; Hu, J.; Li, P. Purification and characterization of plantaricin LPL-1, a novel class IIa bacteriocin produced by Lactobacillus plantarum LPL-1 isolated from fermented fish. Front Microbiol, 2018, 9: 2276. doi: 10.3389/fmicb.2018.02276.
Point 4: Line 112: Reconsider the word “protease stability” since various group of enzymes, i.e. lipase and amylase, were also tested in this study.
Response 4: Thank you for reminding us. Our original writing did create ambiguities, now we have changed "protease stability" to "enzyme stability". Please see line 114 of the revised manuscript.
Point 5: Line 122: Please indicate the indicator strains tested here.
Response 5:
Many thanks for your advice. We have made the change. The changed contexts read:
Line 129-131: To investigate the antimicrobial spectrum of W. coagulans PL-W, the antimicrobial activity of the crude bacteriocins of W. coagulans PL-W against a range of indicators strains, including food spoilage bacteria and food-borne pathogens(Table 1), was determined by using the pour plate method described by An et al.
Point 6: Line 164: Why L. monocytogenes was selected as an indicator strain at this primary screening step?
Response 6:
We are so grateful for your kind question.
- monocytogenes is the most deadly food-borne pathogen(Rogalla D, Bomar PA. Listeria Monocytogenes. 2022 Jul 4. In: StatPearls [Internet]. Treasure Island (FL): StatPearls Publishing; 2022 Jan–. PMID: 30521259).
We thus selected L. monocytogenes as the indicator bacterium at the first screening step in order to screen for a strain with potential applications in food preservation.
Point 7: Line 170: W. coagulans has been recognized as GRAS from the food safety lists such as EFSA (QPS) and IDF/EFFCA microbial food cultures or not?
Response 7:
We thank the reviewer for raising this question.
- coagulans was previously classified as B. coagulans, which has been reported as safe by the United States Food and Drug Administration (FDA) and the European Union Food Safety Authority (EFSA) and is on the list of Generally Recognized as Safe (GRAS) and Qualified Safety Assumptions (QPS)[1]. In 2020, Canadian researchers conducted a more comprehensive genomic analysis of 300 Bacillus strains to shed light on their evolutionary relationships. It was found that most strains, except Bacillus subtilis and Bacillus cereus, needed to be reclassified. The genus Protobacillus was divided into 17 different genera, and the name Bacillus coagulans of the original genus was changed to Weizmannia coagulans[2].
References are as follows:
[1] EFSA Scientific Opinion on The Maintenance of the List of QPS Biological Agents Intentionally Added to Food and Feed (2013 update) EFSA J. 2013;11:3449. doi: 10.2903/j.efsa.2013.3449.
[2] Gupta R S, Patel S, Saini N, et al. Robust demarcation of 17 distinct Bacillus species clades, proposed as novel Bacillaceae genera, by phylogenomics and comparative genomic analyses: description of Robertmurraya kyonggiensis sp. nov. and proposal for an emended genus Bacillus limiting it only to the members of the Subtilis and Cereus clades of species[J]. International journal of systematic and evolutionary microbiology, 2020, 70(11): 5753-5798.
Point 8: Line 277-282: This data should be also addressed as supplementary materials.
Response 8: We appreciate it very much for this suggestion, and we have done it according to your ideas.
Point 9: Line 284-317: Do the authors think that it is necessary to confirm these probiotic potentials using in vitro assays? I would recommend discussing also on these confirmation issues.
Response 9:
We appreciate the reviewer’s insightful suggestion and agree that it would be useful to demonstrate the probiotic properties of W. coagulans PL-W using in vitro assays. Therefore, we have analyzed the antibiotic sensitivity (line 303-305, Table 4) and hemolysis(line 310-312, Supplementary Figure S4) of W. coagulans PL-W, as supplementary material to explain the probiotic properties of the strain. Since we focus on a genome-wide analysis of W. coagulans PL-W to demonstrate the safety of this strain for food preservative applications, the Vitro assays may not be sufficient in this research. In future studies, we will focus on validating the probiotic potentials of W. coagulans PL-W using in vitro assays.
We would like to thank the referee again for taking the time to review our manuscript.
Reviewer 3 Report
The main idea of the present study is to investigate the complete genome sequencing of Weizmannia coagulans and to assess its safety properties. Both of these issues have been investigated in previous studies (https://doi.org/10.3390/ijms23063135 ; https://doi.org/10.3390/life12091388) and therefore the study does not have much new to offer.
Author Response
Response to Reviewer 3 Comments
Reviewer 3
The main idea of the present study is to investigate the complete genome sequencing of Weizmannia coagulans and to assess its safety properties. Both of these issues have been investigated in previous studies (https://doi.org/10.3390/ijms23063135; https://doi.org/10.3390/life12091388) and therefore the study does not have much new to offer.
Response 1: Thank you for this valuable feedback.
We have carefully read the two articles you have listed. In the first article(https://doi.org/10.3390/ijms23063135), the authors demonstrated the genetic diversity of Weizmannia coagulans through comparative genomics, revealing its potential for biotechnology applications. In the second article(https://doi.org/10.3390/life12091388), the authors have isolated a novel isolated Weizmannia coagulans, assessing its potential as a functional probiotic through in vitro experiments. Our article does have similarities to both of them. However, in this study, we are more focused on analyzing the potential application of W. coagulans PL-W as a food protective culture.
​ Bacteriocins are antimicrobial peptides produced by bacterial ribosomes and have potential alternatives to antibiotics[1]. In this study, we have identified a novel W. coagulans PL-W with excellent antibacterial activity from Mongolian Traditional cheese. In contrast to several reported strains of W. coagulans, the bacteriocins produced by W. coagulans PL-W exhibited inhibitory activity against more foodborne pathogens[2, 3], including Listeria monocytogenes CMCC 54004, Bacillus cereus ATCC 14579, and Staphylococcus aureus ATCC 25923. Meanwhile, the bacteriocins have outstanding stability against pH, temperature, and surfactants, and are sensitive to proteases.
​ The application of bacteriocins to food preservation is mainly through the addition of pure bacteriocins, fermentations containing bacteriocins, or bacteriocin-producing strains[4].​ It is important to note that the strain used in the food industry to produce the bacteriocin must be safe. Therefore, we analyzed the safety of W. coagulans PL-W by complete genome sequencing. ​It has been shown that W. coagulans PL-W is a novel strain without virulence and resistance genes detected, which is different from the other reported W. coagulans[5]. ​At the same time, the presence of clusters of genes involved in bacteriocin synthesis, adhesion-related genes, and genes that contribute to acid and bile tolerance suggest W. coagulans PL-W has the potential as probiotics.
​In summary, the antibacterial spectrum difference with other reported W. coagulans and the genomic properties of W. coagulans PL-W indicate its broad potential as a food protection culture.
We would like to thank the referee again for taking the time to review our manuscript.
Reference
- O'Connor, P.M., et al., Antimicrobials for food and feed; a bacteriocin perspective. Curr Opin Biotechnol, 2020. 61: p. 160-167.
- Fu, L., et al., Preservation of large yellow croaker (Pseudosciaena crocea) by Coagulin L1208, a novel bacteriocin produced by Bacillus coagulans L1208. Int J Food Microbiol, 2018. 266: p. 60-68.
- Sreenadh, M., K.R. Kumar, and S. Nath, In Vitro Evaluation of Weizmannia coagulans Strain LMG S-31876 Isolated from Fermented Rice for Potential Probiotic Properties, Safety Assessment and Technological Properties. Life (Basel), 2022. 12(9).
- Yi, Y., et al., Current status and potentiality of class II bacteriocins from lactic acid bacteria: structure, mode of action and applications in the food industry. Trends in Food Science & Technology, 2022. 120: p. 387-401.
- Aulitto, M., et al., A Comparative Analysis of Weizmannia coagulans Genomes Unravels the Genetic Potential for Biotechnological Applications. Int J Mol Sci, 2022. 23(6).
Round 2
Reviewer 1 Report
1. Line 12-15. The sentence “The crude bacteriocins of W. coagulans PL-W showed antibacterial activity against various foodborne pathogens, including Listeria monocytogenes CMCC 54004, Bacillus cereus ATCC 14579, and Staphylococcus aureus ATCC 25923, and have out-standing stability against pH, temperature, surfactants, and sensitive to protease.” is suggested to divide into two sentence.
2. Since it is described in the manuscript that bacteriocin produced by W. coagulans PL-W is an important component to replace antibiotics for bacteriostasis, the main components of crude substance extracted from W. coagulans PL-W, whether it contains bacteriocin and its content were not detected in the experiments. It is suggested to supplement relevant experiments.
3. Line 351. Please adjust the title of subheading 3.7.
4. There are many grammar and logical problems that are difficult to understand. The authors are suggested to go through the whole manuscript again, and check for potential typos, grammar and logic problems.
Author Response
Dear reviewer:
Thank you for your decision and constructive comments on our manuscript. We have carefully considered your suggestion and made some changes. We have tried our best to improve and made some changes to the manuscript.
We are grateful for your comments and have addressed them as follows:
Point 1: Line 12-15. The sentence “The crude bacteriocins of W. coagulans PL-W showed antibacterial activity against various foodborne pathogens, including Listeria monocytogenes CMCC 54004, Bacillus cereus ATCC 14579, and Staphylococcus aureus ATCC 25923, and have out-standing stability against pH, temperature, surfactants, and sensitive to protease.” is suggested to divide into two sentence.
Response 1:
Many thanks for your suggestions, the text has been modified, please see lines 12-15.
The contexts read:
The crude bacteriocins of W. coagulans PL-W showed antibacterial activity against various foodborne pathogens, including Listeria monocytogenes CMCC 54004, Bacillus cereus ATCC 14579, and Staphylococcus aureus ATCC 25923. Moreover, the crude bacteriocins have outstanding stability against pH, temperature, surfactants, and are sensitive to protease.
Point 2: Since it is described in the manuscript that bacteriocin produced by W. coagulans PL-W is an important component to replace antibiotics for bacteriostasis, the main components of crude substance extracted from W. coagulans PL-W, whether it contains bacteriocin and its content were not detected in the experiments. It is suggested to supplement relevant experiments.
Response 2:
Many thanks for your good advice, according to your suggestion, we supplement experiments about purification of the bacteriocins, and concluded that the bacteriocin extracted from W. coagulans PL-W was Circular A, the entire amino acid sequence was MGLFHVASKFHVSAGIASGVVTAVLHAGTIASIIGAVTVVMSGGVDAILDMGWTAFIAEVKHLAKEYGKKRAIAW.
please see lines 139-150, lines 261-270, and lines 399-412.
The context read:
lines 139-150,
2.3.4 Purification of Bacteriocin
To identify bacteriocin, firstly, ultrafiltration tubes of 10KD and 3KD were used to isolate the crude bacteriocin. The fractions containing proteins larger than 10KD, between 3KD-10KD, and smaller than 3KD were tested for antibacterial activity. Then, the active fractions were loaded onto a C18 reverse-phase column (5 µm, 4.6 × 250 mm, Agilent, CA, United States), connected to a reverse-phase high-performance liquid chromatography (RP-HPLC) system, and eluted at 0.5 mL/min flow rate by a liner gradient elution with 95% water-acetonitrile (5–95%) containing 0.1% trifluoroacetic acid (TFA) in 30 min. The different peaks were collected at an absorbance was 280 nm, then concentrated using 1KD ultrafiltration tubes for antibacterial activity evaluation. Using Tricine-SDS-PAGE(16.5% separated and 4% concentrated gel) analyze the range of molecular mass of the collected active fractions.
lines 261-270
3.2.4 Purification of Bacteriocin
The Crude antimicrobial substances were separated into different fractions using 10KD and 3KD ultrafiltration tubes. The result showed that only fractions with mo-lecular mass between 3-10 kDa exhibited antibacterial activity(Figure 3A). Therefore, it was presumed that the molecular mass of the bacteriocin was among 3-10 kDa. Fur-ther purification of the active fraction using a C18 column showed that only the third peak was able to inhibit the growth of the indicator strain (Figure 3B). Tricine-SDS-PAGE analysis of the molecular mass of peak 3 displayed a single band around 7 kDa(Figure 3C). Consequently, we conclude that W. coagulans PL-W expressed a bac-teriocin with a molecular mass near 7 kDa.
lines 399-412
Tricine-SDS-PAGE analysis of the bacteriocins indicated that its molecular mass was near 7 kDa(Figure 3C), combing the genomic information, we concluded the bac-teriocin extracted from W. coagulans PL-W was Circular A, the entire amino acid sequence was MGLFHVASKFHVSAGIASGVVTAVLHAGTIASIIGAVTVVMSGGVDAILDMGWTAFIAEVKHLAKEYGKKRAIAW. Although the genome was con-firmed for the presence of the gene of Amylocyclicin, purification of crude bacteriocin did not find it. Thus, we speculate that Amylocyclicin may be expressed only under certain specific growth conditions. Or Amylocyclicin did not show inhibitory activity against the L. monocytogenes CMCC 54004 that we used in the purification process. In this regard, it may be possible to use heterologous expression methods in the future to detect the antimicrobial ability of Amylocyclicin or attempt to use other bacteria ra-ther than L. monocytogenes as indicator strain when purifying the bacteriocins. In gen-eral, the expression of Circularin A may provide a competitive advantage for W. coag-ulans PL-W.
Point 3: Line 351. Please adjust the title of subheading 3.7.
Response 3:
Many thanks for your good advice.
​We have changed the title of subheading 3.7 from “Antimicrobial compound genes prediction “to “Antimicrobial compound genes prediction and validation”. Please see line 373.
Point 4: There are many grammar and logical problems that are difficult to understand. The authors are suggested to go through the whole manuscript again, and check for potential typos, grammar and logic problems.
Response 4: We regret there were problems with English. We have carefully revised the whole manuscript again and uploaded a clean version for reading.
We would like to thank you again for taking the time to review our manuscript.
Reviewer 3 Report
---
Author Response
Thank you very much for giving us the opportunity to revise this article. According to your suggestion, we have strengthened the introduction. Please see lines 65-79.
The contexts read:
It is noteworthy that the strains used to produce bacteriocin must be safe. Thus, assessing the safety and probiotic properties of the strains is an essential step for use in food products [23, 24]. For instance, Sreenadh et al. have evaluated the probiotics, safety, and technology of W. coagulans S-31876 through a number of in vitro experiments to explore its potential applications[25]. High-throughput sequencing enables the evaluation of the properties of strains at the genomic level, including their genetic, safety, and metabolic profiles. Previously, Aulitto et al. used comparative genomics to focus on the biotransformation and defense ability of W. coagulans against the external environment[26]. In this research, the W. coagulans PL-W with antibacterial activity was identified from Mongolian Traditional cheese, and its complete genome was confirmed. Assessment of the genome indicates that W. coagulans PL-W may be a safe strain with probiotic properties to use and promote future research and development of the organism in food preservation. In addition, the crude bacteriocin characteristics produced by W. coagulans PL-W were evaluated to provide the theoretical basis for its potential application as food preservative.